# Improving human activity classification based on micro-doppler signatures of FMCW radar with the effect of noise

NgocBinh Nguyen [1], MinhNghia Pham [1]*, Van-Sang Doan[2], VanNhu Le[1]

**1** Faculty of Radio Electronics Engineering, Le Quy Don Technical University, Hanoi, Vietnam, **2** VietNam Naval Academy, Nha Trang, Khanh Hoa, Vietnam

* nghiapmmta@gmail.com

**Data Availability Statement:** All Our data files are available from the https://doi.org/10.6084/m9. figshare.25874515.v1 database.

## Abstract

Nowadays, classifying human activities is applied in many essential fields, such as healthcare, security monitoring, and search and rescue missions. Radar sensor-based human activity classification is regarded as a superior approach in comparison to other techniques, such as visual perception-based methodologies and wearable gadgets. However, noise usually exists throughout the process of extracting raw radar signals, decreasing the quality and reliability of the extracted features. This paper presents a novel method for removing white Gaussian noise from raw radar signals using a denoising algorithm before classifying human activities using a deep convolutional neural network (DCNN). Specifically, the denoising algorithm is used as a preprocessing step to remove white Gaussian noise from the input raw radar signal. After that, a lightweight Cross-Residual Convolutional Neural Network (CRCNN) with adaptable cross-residual connections is suggested for classification. The analysis results show that the denoising algorithm with a range-bin interval of 3 and a cut-threshold value of 3 achieves the best denoising effect. When the denoising algorithm was applied to the dataset, CRCNN improved the right classification rate by up to 10% compared to the recognition results achieved with the original noise-added dataset. Additionally, a comparison of the CRCNN with the denoising algorithm solution with six cutting-edge DCNNs was conducted. The experimental results reveal that the proposed model greatly outperforms the others.

## Introduction

The precise identification and categorization of daily human actions has been applied extensively across various domains, ranging from civilian life to national defense and security, which consists of healthcare, border surveillance, and search and rescue operations. The recent interests in healthcare are on the rise because the rising aging rate of the population has led to a concurrent increase in the risk of falls and fall-related injuries among the elderly and people with special needs [1]. Hence, it is essential to precisely categorize indoor living human actions to rapidly alert the potentially hazardous incidents such as falls and strokes. Several health-

**Funding:** The author(s) received no specific funding for this work.

**Competing interests:** The authors have declared that no competing interests exist.

monitoring gadgets are available to address this issue, such as vision-based techniques [2], wearable devices [3], and radar sensors [4]. In good lighting conditions, vision-based techniques can work well with high accuracy and without affecting user comfort. However, it is greatly affected by surrounding lighting and weather conditions and violates the user's personal privacy [5]. Although wearable devices can completely overcome the disadvantages of vision-based techniques, their most significant drawback is that they must be worn continuously, causing certain discomfort to users, especially when resting or going to bed. Out of all the technologies mentioned, radar stands out due to its superiority in various aspects, as seen in Table 1. Therefore, the utilization of radar systems for human activity recognition is a reasonable choice due to its benefits in terms of personal privacy compliance, non-contact sensing, and being unaffected by weather conditions. Doppler radar is a potential candidate for the above-mentioned problem thanks to its ability to depict the additional frequency modulation of micro-motions induced by the limbs in the primary Doppler frequency shift caused by entire body movements. Each activity offers a unique m-D signature, represented in spectrograms, which are used to extract features for identifying human activities [6].

In the recently reported publications, improving the classification accuracy of activities based on the m-D signatures of spectrograms often focuses on two main directions. The first direction is to pay attention to the preprocessing step from the input signal to improve the quality of the spectrogram image before being utilized as input for classifiers. In [7, 8], a proposed technique using pattern contour-confined Doppler-time (PCC-DT) maps can minimize redundant information and remove outlier spots to boost the ability to recognize and classify falls. The experimental findings reveal that the proposed approach can identify sudden and soft fall motions with high accuracy. In [9], the C-MCA-STFrFT algorithm is used to classify motions of human activities with small or limited limb movements by separating the torso and limb components. The experimental results show that the accuracy improved at least up to 5%. In [10], a novel method that employs a 'mask' to emphasize the area of usefulness from the m-D signature is called adaptive thresholding. These masked signatures on the spectrogram serve to extract features and enhance accuracy. As a result, the right recognition approaches 92.5% for six activities. In [11], to increase the accuracy of identifying daily human activities, the authors applied one more preprocessing step, which is called the ECM-Th-STFT separation method, to separate the m-D signatures corresponding to the limbs from the Doppler signal of the torso. With a selected threshold value of 0.3, the ECM-Th-STFT method has improved accuracy by up to 6% over the initial unseparated dataset.

The second branch relies on the automatic feature extraction ability of Convolutional Neural Networks (CNNs) to improve accuracy in performing the task of monitoring and classifying activities. In [12], AlexNet, VGGNet, and a self-designed CNN are used to extract and classify human actions. With an accuracy of about 95%, VGGNet has a higher recognition rate than others, but it comes at the cost of longer processing times and more complexity. Besides using existing CNN models, an approach that involves adjusting the conventional CNN design, developing lightweight models with simple forms, and designing improved models using up-to-date methodologies, known as self-custom design models, has developed as a

**Table 1. Comparison between various health-monitoring methods.**

| Gadgets | Lighting independent | Weather independent | Privacy protection | Comfortable |
|---|---|---|---|---|
| Vision-based methods | No | No | No | Yes |
| Wearable devices | Yes | Yes | Yes | No |
| Radar | Yes | Yes | Yes | Yes |

novel approach for researchers in recent years. Proposals [12–14] are the studies in this branch. With a simplistic architecture that only consists of a few convolutional layers linked to activation layers via straightforward connections [12, 13], effectively solve the feature extraction of human activities via the input spectrograms; however, the accuracy of these models is not high, at just 91.35% and 90.3%, respectively. In [14], a Dense Inception Neural Network (DINN) updated the architectures of Inception modules and leveraged shortcut connections from DenseNet to handle the problem of gradient vanishing that results from the layering of sequential convolution and activation layers in a deep CNN. The experiment indicated that the self-custom design-based methodology provides the optimal balance in terms of accurate classification, model size, and prediction time compared to the existing CNN-based method.

Both of the above research directions have effectively improved classification accuracy. However, significant challenges remain for radar data processing, including the effective classification of varied human activities in real environments with noise effects. Noise exists throughout the process of receiving radar signals, decreasing the quality and reliability of extracted features from radar signals [15, 16]. Under the influence of noise, the extracted features to classify these actions are greatly submerged in the background, which leads to a significant decline in classification accuracy. Therefore, denoising techniques play an important role in improving the classification accuracy of human activities from radar signals.

Various denoising methods have been developed for the radar signal, such as wavelet denoising, threshold filter, and adaptive threshold filter [17–19]. Besides, denoising in machine vision has made rapid progress [20], there is also extensive research focused on signal denoising obtained from vision through deep neural networks. Model-based denoising methods address the issue of additional white Gaussian noise using designs like U-Net [21] and residual learning [22]. Model-based denoising methods have had significant noise reduction results, however, the input noisy images are mainly digital images, the denoising of input signals as complex signals (In-Phase and Quadrature—I & Q) is still limited. Moreover, the importance of signal pre-processing has been almost forgotten due to too much reliance on machine learning to find patterns for recognition. Preprocessing methods, especially denoising techniques, need to be introduced before the classification model with the aim of improving the reliability and accuracy of the classification. Therefore, the major research in this study will focus on both directions, including a proposed algorithm to eliminate white Gauss noise from the noisy raw data at the preprocessing step and a novel lightweight self-designed network to reduce computational complexity and improve classification accuracy. The key contributions of this paper can be summarized as follows:

Firstly, propose a denoising algorithm accomplished through two primary steps: determine the optimal range-bin interval using minimum entropy information [23–25], then identify an appropriate cut-threshold value through experimentation to remove white Gauss noise from the input noising signal.

Secondly, a novel customized lightweight CNN, namely Cross-Residual Convolutional Network (CRCNN), is proposed for classifying indoor human activities based on m-D signatures obtained from the denoised spectrograms. The three main innovations of the proposed model show as follows:

- These filters of CRCNN have different sizes, including $1 \times 1$, $3 \times 3$, and $5 \times 5$, which are intelligently designed to be able to extract the most diverse features from previous feature maps, including global and local distribution.

- Cross-residual connections are employed appropriately in im-res blocks to reuse previously useful feature maps that may be lost when going through the model's extraction core flow.

- The classification accuracy of CRCNN outperforms six existing state-of-the-art DCNNs and is validated by comparing it with RepVGG [26], Dop-DenseNet [27], MobileNet-V2 [28], ResNet [29], DINN [14], and DIAT-RadHARNet(DIAT) [30] with varying signal-to-noise ratios to demonstrate its robustness.

Finally, the proposed approach is evaluated and compared with several existing denoising methods. The results demonstrate that the average classification accuracy improves up to 10% compared to the original noisy dataset, in which the accuracy of the proposed CRCNN model is also higher than the referencing models and surpasses 99% at a -10 dB noise level.

The rest of this paper is organized as follows: The FMCW radar and data preparation section introduces an overview of FMCW radar and data preparation. The proposed denoising algorithm and the suggested CRCNN are presented in the proposed method section. The experimental results and discussion section discusses experimental and comparative results. The last section is the conclusion.

## FMCW radar and data preparation

### Overview of the FMCW radar

The block diagram of a basic frequency-modulated continuous wave (FMCW) radar system consisting of one transmitter and one receiver is detailed in Fig 1a. FMCW transmits a sinusoid with linearly modulated frequencies over time. There are various types of linear frequency modulation, including sawtooth, triangle, or linear segment, of which the most basic and widely used is sawtooth modulation, which involves a succession of chirps over time (as seen in Fig 1b).

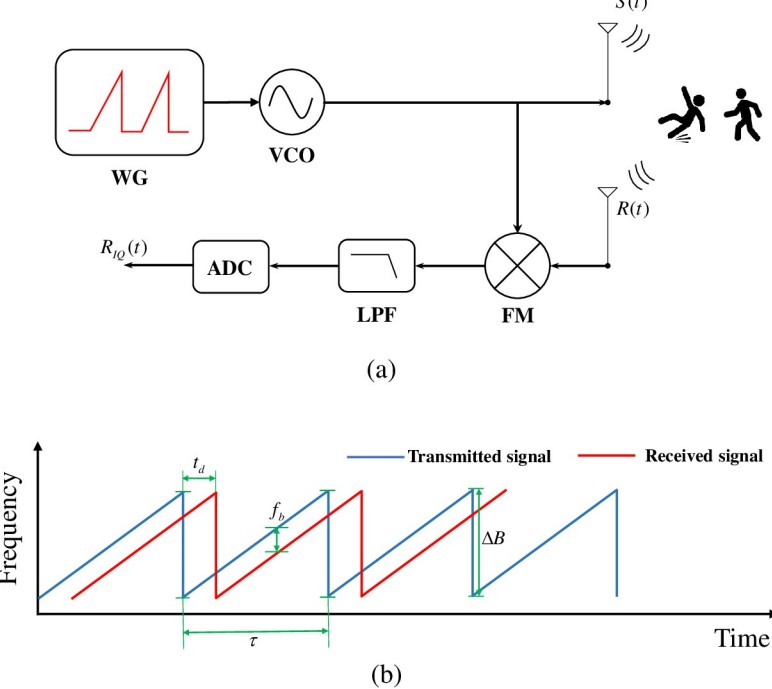

**Fig 1. Overview of the FMCW radar: (a) The FMCW radar block diagram, (b) FMCW sawtooth waveform.**

The transmitted signal $S(t)$ is defined as [31]:

$$S(t) = A_t \cos\left(2\pi\left(f_0 t + \frac{\Delta B}{2\tau}t^2\right)\right) \quad (1)$$

where, $\Delta B$ is the sweep bandwidth, $\tau$ is the chirp duration, $A_t$ is the amplitude and $f_0$ is the carrier frequency of the transmitted signal. The reflected signal when hit the target is shown:

$$R(t) = A_r \cos\left(2\pi\left(f_0(t - t_d) + \frac{\Delta B}{2\tau}(t - t_d)^2\right)\right) \quad (2)$$

where, $t_d = \frac{2D}{c}$ is the round-trip delay. $D$ is the distance to the object, and $c = 3 * 10^8 m/s$ is the speed of light.

The IF signal, also known as the beat signal, following the mixer is the result of multiplying the receive and transmit signals and can be defined by:

$$R_{IQ}(t) = A_m \exp\{2\pi k t_d t - \pi k t_d^2 + 2\pi f_c t_d\} = A_m \exp\{\varphi(t)\} \quad (3)$$

in which, $k = \frac{\Delta B}{\tau}$, $A_m = \frac{A_t A_r}{2}$, and $\varphi(t)$ is the phase of the beat signal.

Instantaneous frequency is a derivative of the phase of the beat signal:

$$f_b = \frac{1}{2\pi}\frac{d\varphi(t)}{dt} = k t_d = \frac{\Delta B}{\tau}\frac{2D}{c} \quad (4)$$

Fig 2 shows the FMCW processing flow from the IF signal. The data matrix is obtained after the IF signal passes through the ADC block, which can be used to calculate the range profile

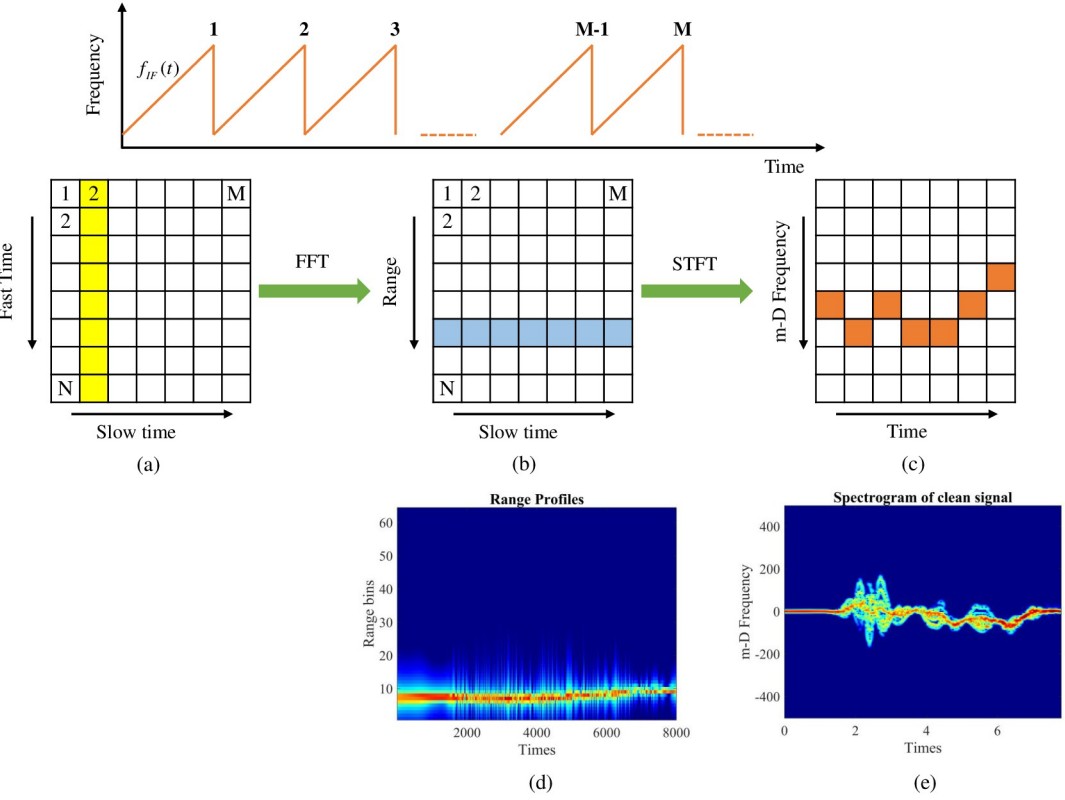

**Fig 2. FMCW processing flow from the IF signal.**

and time-frequency distribution of the target [32]. The data matrix, $D \in \mathbb{C}^{N \times M}$, each column stores data of 1 chirp, $x[j, i]$, $j \in (1, N)$ and $i \in (1, M)$, as shown in Fig 2a.

The first fast Fourier transform algorithm (FFT) is directly applied for each chirp along the fast time axis, while the range bins are stored in the range FFT matrix (Fig 2b). The FFT representation at $i^{th}$ chirp ($i = 1, .., M$) with samples is defined as follows:

$$X_i[k] = \sum_{n=0}^{N-1} x_i[n] \exp\left(-j\frac{2\pi}{N}kn\right) \tag{5}$$

where, $X_i[k]$ is the discrete Fourier transform of the $i^{th}$ chirp at $k^{th}$ frequency bin, and $x_i[n]$ is the value of the $i^{th}$ chirp at the $n^{th}$ sample. Perform FFT sequentially from 1 to $M$ chirps to obtain the range profile (Fig 2b). This first FFT determines the presence of the target at different distances, which is called range-FFT. As a result, the range profile of the corresponding target could be obtained (Fig 2d).

When the short-time Fourier transform (STFT) is employed on the chirp indexes along the slow-time axis on all the range-FFT bins, the m-D of the target is found (Fig 2c). STFT is employed on the chirp indexes along the slow-time axis on all the range-FFT bins is shown as 6:

$$STFT = \sum_{j=1}^{N} STFT\{X_j[m]\} \tag{6}$$

$$STFT\{X_j[m]\} = \chi_j(l, m) = \sum_{m=lH}^{lH+S-1} X_j[m]w[m - lH] \exp\left(-j\frac{2\pi\omega m}{S}\right) \tag{7}$$

where, $STFT\{X_j[m]\}$ is STFT, which is applied on the $j^{th}$ range-bin indexes along the slow-time axis, $H$ is the size of the step between consecutive windows, $S$ is the size of the window (number of samples in each window), $w[m]$ is the window function applied to each window, and $l \in [0:L]$ is the index of the STFT window, with $L = \lfloor \frac{M-S}{H} \rfloor$.

The acquired result is the spectrogram containing m-D signatures corresponding to various activities of the interested target (Fig 2e). The spectrogram representations of the first FFT and second STFT are shown in Fig 2d and 2e, respectively. Fig 2d shows the range of the target during the movement, which is used to determine the target's location; however, this information is insufficient to classify human activities. Fig 2e illustrates the m-D frequency shift versus time, which provides insight into the movements of various body components, including the torso and limbs, over time. These signatures are unique to different activities and are employed to categorize human activities. Therefore, the spectrogram image depicted in Fig 2e is utilized as input for the classifiers in this study.

## Data preparation

The dataset generated by Simhumalator software [33] consists of 11 daily human activities: walking, body rotation, punching, kicking, grabbing, standing up, sitting down, standing to walk, walking to sit, walking to fall, and falling to walk. The data is collected using FMCW radar with a carrier frequency of 24 GHz (K-band). Each chirp has a bandwidth of 400 MHz and a duration of 1 ms. The raw signal is sampled at a sampling rate of 128 samples per chirp. The radar is arranged 1m above the ground, and the distance from the radar to the objects is 3m. A total of eleven distinct actions are performed and repeated 60 times with various aspect angles [0˚, -45˚, 45˚, -90˚, 90˚], respectively. The time for each execution is 8–15 seconds. In addition, white Gauss noise is added to the dataset with signal-to-noise ratio (SNR) values

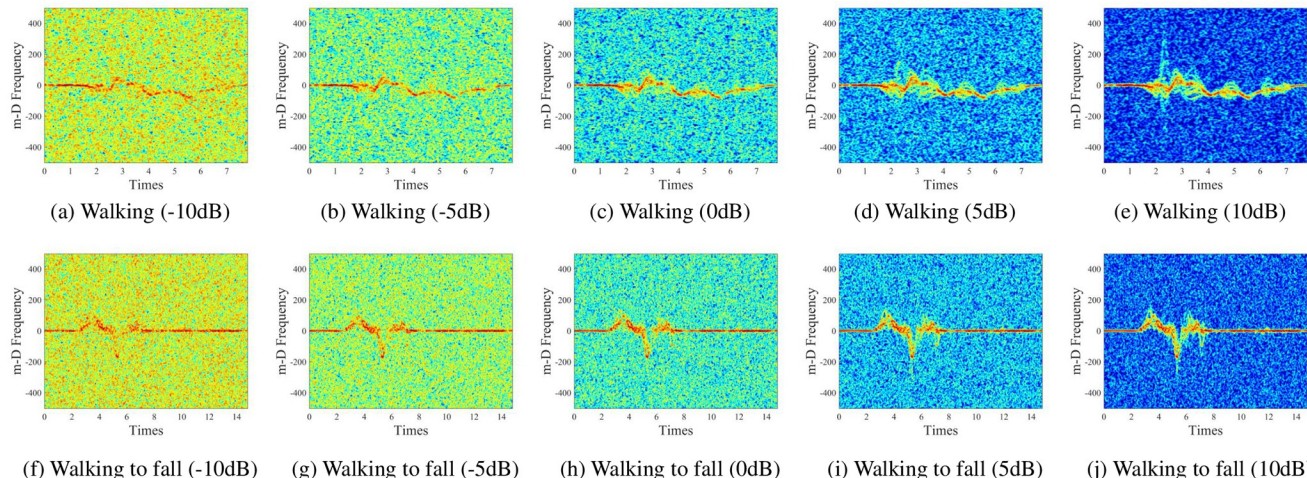

(a) Walking (-10dB)  (b) Walking (-5dB)  (c) Walking (0dB)  (d) Walking (5dB)  (e) Walking (10dB)

(f) Walking to fall (-10dB)  (g) Walking to fall (-5dB)  (h) Walking to fall (0dB)  (i) Walking to fall (5dB)  (j) Walking to fall (10dB)

**Fig 3. The spectrogram of walking and walking to fall actions in the noise-added dataset with different SNR levels.**

ranging from -15 to 10 dB to make it more like real classification conditions and enhance the challenge of the model. As a result, the raw signal dataset includes 19800 samples, which are obtained by combining 11 activities, 5 aspect angles, 60 iterations, and 6 noise levels.

The spectrogram of walking and walking to fall actions in the noise-added dataset with different SNR levels is shown in Fig 3, which illustrates that different actions have unique m-D signatures and features that are mostly obscured by the background noise, rendering them challenging to detect and differentiate under conditions of low noise levels.

## Proposed method

The overall diagram of the proposed approach is given in Fig 4. The Denoising and Processing block is responsible for removing white Gauss noise from the noisy raw data. The output of this block is the denoised spectrograms before being fed into the classifier. Then, a lightweight DCNN model with a design based on cross-residual connections, namely the Cross-Residual Convolutional Neural Network (CRCNN), is used to classify human activities based on m-D signatures obtained from the input denoised spectrograms.

### Proposed denoising approach

It is evident from Fig 3 that with lower noise levels, the m-D signatures are almost completely submerged under the noise background and cannot be observed. It leads to data-driven

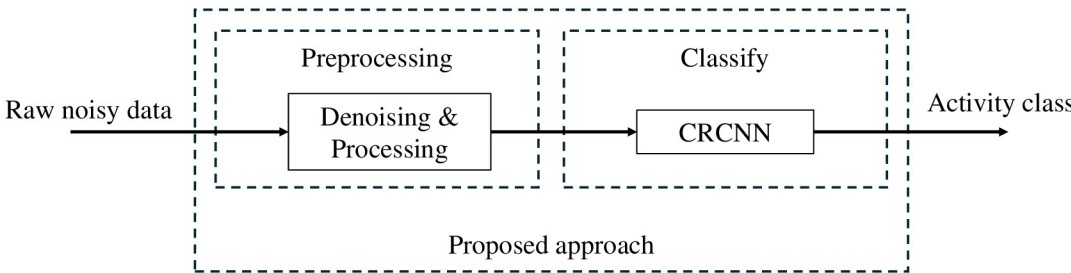

**Fig 4. The diagram of the proposed approach.**

automatic feature extraction utilizing DCNNs being limited, and the categorization accuracy of activities will be low. Therefore, to enhance the extracted features by DCNNs, the m-D signatures on the acquired spectrogram must be denoised and made more prominent. A new approach consisting of two steps for denoising is proposed to resolve this problem, including:

Step 1: Determine the optimal range-bin interval based on the minimum information entropy value.

Step 2: Determine the optimal cut-threshold value.

**Determine the optimal range-bin interval based on the minimum information entropy value.** • Minimum entropy criterion

Information entropy, which measures data uncertainty, is a cornerstone of information theory. Higher uncertainty corresponds to higher entropy, while lower uncertainty results in lower entropy levels. In the context of time-frequency distributions, entropy assesses the uncertainty inherent in the frequency distribution at a specific time. For a discrete probability distribution $p = (p_1, p_2, \cdots, p_n)$, its entropy is calculated as $H(p) = -\sum_i p_i \ln p_i$, where $p_i$ represents the probability of each value. Similarly, in the time-frequency domain, if $p = (p_1, p_2, \cdots, p_F)$ represents the frequency distribution at a certain time t, with the entropy function $H_t(p_t) = -\sum_F p_t \ln p_t$ quantifies the uncertainty in the frequency distribution. The entropy function exhibits symmetry and extremum properties. The first property shows that the information entropy remains unchanged when the order of variables in the probability distribution sequence is altered. The second property shows that the information entropy reaches maximum entropy when the energy distribution of frequency resolution units is uniform and minimum when only one frequency unit dominates. This is also an important basis for the algorithm to track the energy peak by using the minimum information entropy [23–25].

The time-frequency distribution in 6 can be interpreted as a probability distribution sequence [23]. The probability distribution of frequency resolution unit at time $\tau$ can be transformed into their discrete forms as

$$p_\tau(\omega) = \frac{\left|\chi_{range_{r_q}}(\tau, \omega)\right|^2}{\sum\limits_{\omega=1}^{W}\left|\chi_{range_{r_q}}(\tau, \omega)\right|^2} \tag{8}$$

$$H_\tau = -\sum_{\omega=1}^{W} p_\tau(\omega) \cdot \ln p_\tau(\omega) \tag{9}$$

in which $\chi_{range_{r_q}}(\tau, \omega)$ is the STFT of the input signal, $X_{range_{r_q}}[m]$, with the selected range-bin interval is determined by $r_q$, instead of STFT on all range-bin indexes, as in 6. $H_\tau$ is the information entropy of the frequency distribution at time $\tau$ following STFT. The acquired average information entropy value at all times over the entire signal length, $T$, is represented

$$H_{avg}(range_{r_q}) = \frac{\sum\limits_{\tau=1}^{T} H_\tau}{T} \tag{10}$$

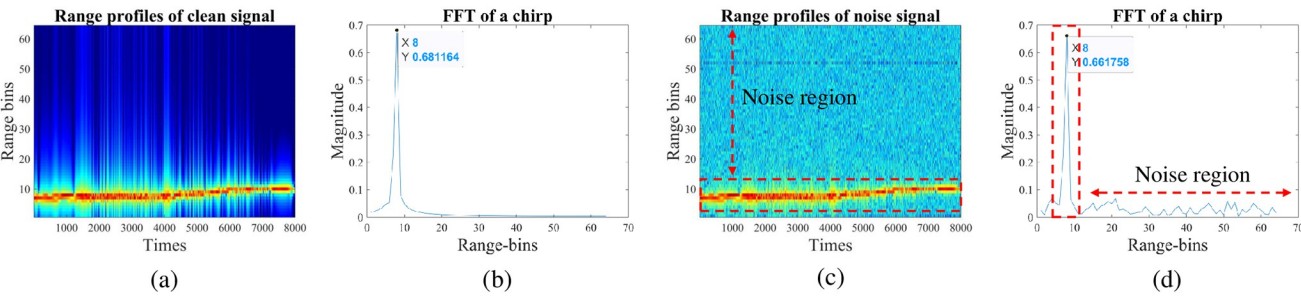

**Fig 5. Range profile and FFT of a specific chirp.**

The magnitude of $H_{avg}(range_{r_q})$ can reflect the average energy distribution of the signal in the time-frequency domain. By searching for the minimum information entropy, it is possible to determine the optimal range-bin region to achieve the highest average energy distribution over the entire signal. Therefore, to obtain the best time-frequency distribution, the criterion of minimum information entropy is used for STFT by finding the minimum value defined by:

$$r_{opt} = \arg \min_{1 \le q \le Q} H_{avg}(range_{r_q}) \tag{11}$$

where, $H_{avg}(range_{r_q})$ is the information entropy corresponding to the $q^{th}$ element of the range-bin library with $Q$ elements. 11 is the criterion for minimum information entropy.

• Determine the optimal range-bin interval

As mentioned in the FMCW radar and data preparation section, the m-D signatures of the target are obtained using STFT on all range bins in the slow-time direction. However, in the case of noise-added signals, selecting optimal range-bin intervals to extract m-D signals is one of the effective noise reduction methods. Fig 5a and 5c depicts the range profile of walking action with clean data and data with a SNR of 5 dB. In both cases, the target is detected at $8^{th}$ range-bin and fluctuates between $6^{th}$ and $10^{th}$ range-bins during the movement. Fig 5b and 5d displays the FFT of a specific range-bin index corresponding to clean data and noise-added data with a 5 dB-SNR, respectively. Observing more detail, Fig 5d indicates that the highest concentrated energy level indicating the target similarly appears in the $8^{th}$ range-bin (identical to Fig 5b), and energy peaks outside the red dashed area are considered noise. In the following section, therefore, the authors will utilize the minimum entropy value to determine the optimal range-bin index to eliminate undesirable noise components.

The complete procedure of determining the optimal range-bin interval is summarized in Algorithm 1.

**algorithm 1** Find optimal range-bin interval:

```
1: Initialization: Data matrix, range-bin intervals library r = {r₁,
   r₂, ..., r_Q} with r_q, q = 1, 2, ..., Q.
2: for i = 1 : M do
3:   Calculate X_f = FFT{x[:, i]} (according to 5)
4:   Find P_max[i] = max{abs(X_f)}
5:   Find index of P_max[i]
6: end for
7: Determine P_max corresponding with M elements
8: Find the idx_max is the most repeated index in P_max
9: for q = 1 : Q do
10:   Determine range_r_q
```

```
11:    Calculate STFT along slow-time axis with range_r_q
12:    Calculate H_avg(range_r_q) (according to 10)
13: end for
14: Determine H_avg corresponding with Q elements
15: Find r_opt (according to 11)
16: range_opt = (idx_max - r_opt : idx_max + r_opt)
17: Output: range_opt.
```

At $i^{th}$ chirp, after performing the first FFT, we will determine the range-bin index has the highest magnitude value, $P_{\max}(i)$. The $\mathbf{P_{max}}$ is a vector including elements containing the range-bin indexes with the highest magnitude value after FFT of all chirps. After that, determine the $idx_{\max}$ is the most repeated index in $\mathbf{P_{max}}$.

Next, $\mathbf{r} = \{r_1, r_2, \ldots, r_Q\}$ is a series of values of the selected range-bin intervals for consideration (these values are chosen based on experience). Then, $\mathbf{r}$ becomes the library of the range-bin index for the STFT algorithm, and each element in the range-bin library $r_q, q = 1, 2, \ldots, Q$ is selected in turn to perform the STFT. Corresponding to each value of $r_q$, the selected range-bin interval is: $range_{r_q} = [idx_{\max} - r_q : idx_{\max} + r_q]$.

The obtained STFT value for the selected range-bin region corresponding to $q^{th}$ value in the $\mathbf{r}$ is defined:

$$STFT\{X_{range_{r_q}}[m]\} = \sum_{idx_{\max}-r_q}^{idx_{\max}+r_q} STFT\{X_{idx_{\max}}[m]\} \tag{12}$$

The $H_{avg}(range_{r_q})$ value of $STFT\{X_{range_{r_q}}[m]\}$ is calculated corresponding to each $q^{th}$ value in the range-bin library according to 10. Then, $r_{opt}$ is calculated by 11. The selected optimal range-bin interval is $range_{opt} = (idx_{\max} - r_{opt} : idx_{\max} + r_{opt})$.

**Determine a cut-threshold value.** The complete procedure of determining an appropriate cut-threshold value is summarized in Algorithm 2

**algorithm 2** Find the cut-threshold value:

```
1: Initialization: FFT matrix, range_opt, window function (type: Gauss-
   ian, size: 128, overlap: 90%).
2: for l = 1 : L do
3:    Calculate FFT at each frame F_j(l, ω)
4:    Calculate avg(F_j(l, ω))
5:    Design the masking function T_j(l, ω)
6:    Calculate F_j(l, ω)_denoise = F_j(l, ω) ⊙ T_j(l, ω)
7: end for
8: Calculate STFT_denoise{X_j[m]}
9: Calculate STFT_denoise based on range_opt
10: Output: STFT_denoise.
```

STFT is applied on the $j^{th}$ range-bin indexes along the slow-time axis is displayed as 7. Therefore, the Fourier transform of the product between $X_j[m]$ and window function $w[m]$ at any the index of the STFT window, $l$, as follow:

$$F_j(l, \omega) = X_j[m]w[m - lS] \exp\left(-j\frac{2\pi\omega m}{L}\right) \tag{13}$$

Fig 6a shows the results of $F_j(l, \omega)$ at any index of the STFT window, $l$, on the slow-time axis. Obviously, the concentrated energy areas are decided in the central region (this region is to determine the activities of the torso and corresponding limbs). The region outside the red dashed area is considered noise. The noise reduction processes are carried out as follows:

First, calculate the mean value of $F_j(l, \omega)$ is $avg(F_j(l, \omega))$

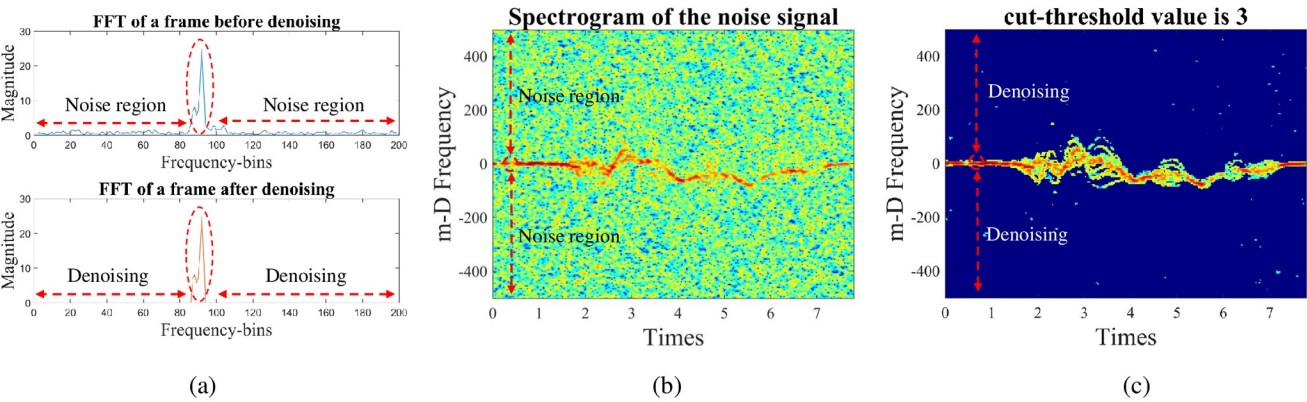

**Fig 6. The FFT representation at one frame in the STFT processing and the denoising result at -5-dB SNR.**

Second, design the masking function $T_j(l, \omega)$ as follow

$$T_j(l, \omega) = \begin{cases} 1, & |F_j(l, \omega)| > ETh \\ 0, & \text{others} \end{cases} \tag{14}$$

with $Eth = Th \times avg(F_j(l, \omega))$, where $Th$ is the cut-threshold that is determined empirically.

Next, $F_j(l, \omega)_{denoise} = F_j(l, \omega) \odot T_j(l, \omega)$, in here, $\odot$ represents element-wise multiplication.

Then, continuously repeat the above steps until the entire length of the signal to calculate $STFT_{denoise}\{X_j[m]\}$ according to 7.

Finally, $STFT_{\text{denoise}}$ is claculate as follows:

$$STFT_{denoise} = \sum_{idx_{\max} - r_q}^{idx_{\max} + r_q} STFT_{denoise}\{X_j[m]\} \tag{15}$$

Fig 6b shows the spectrogram of walking action at a -5 dB-SNR, and Fig 6c is the spectrogram after denoising. Evidently, the noise has been significantly less.

## Proposed CRCNN-based classification approach

A lightweight DCNN model with a design based on cross-residual connections, namely the Cross-Residual Convolutional Neural Network (CRCNN), is proposed to classify human activities based on m-D signatures. The overview diagram of CRCNN is detailed in Fig 7.

The overall architecture of the CRCNN includes three main parts: an input block, the main feature extraction block "Cross-Residual" and an output block. The input block has a size equal to the size of the input spectrogram. The main feature extraction block, called "Cross-Residual" is made up of six C-R connections and seven im-res blocks (shown in Fig 7). The black line shows the straight propagation, and the red line shows the cross-connection. The output block, which contains a Conv-unit-2, a fully connected (FCN) layer, a softmax layer, and a classification (class) layer, is used to classify the action classes.

The CRCNN starts with the input block, which is arranged in a sequential connection with an input layer and a conv-unit-1. To be more specific, the size of the input layer is identical to the size of the input spectrogram images. The Conv-unit-1 is composed of three consecutive layers, including a convolutional layer (conv), a maxpooling layer (maxpool), and the rectified linear unit (ReLU) function, respectively. For detailed structure, the convolutional layer is

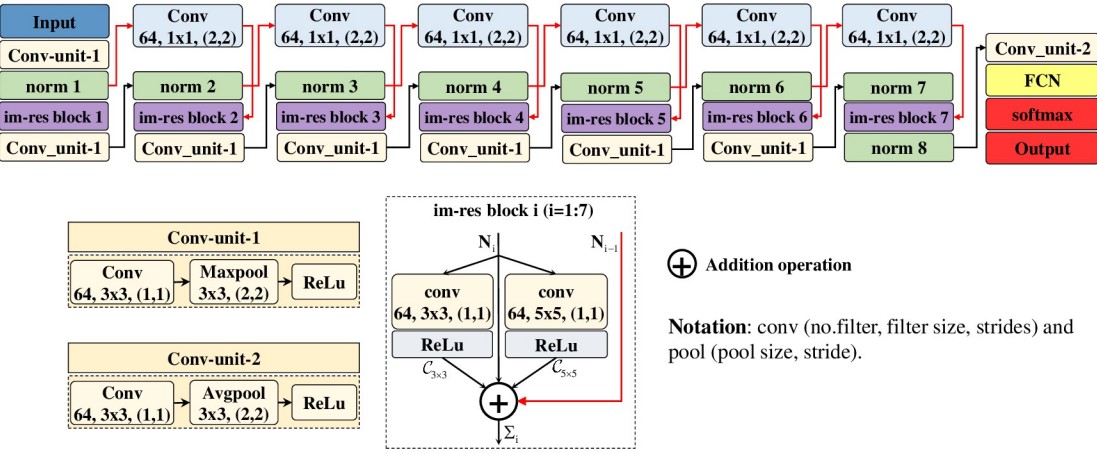

**Fig 7. The overall architecture of the proposed CRCNN.**

fixed with 64 filters to create 64 feature maps correspondingly. The operation of convolution is given as follows:

$$\mathbf{y}(i) = conv(\hat{\mathbf{x}}, \mathbf{c}) = \sum_i \hat{\mathbf{x}}(i)\mathbf{c}(i) + b \tag{16}$$

where $\hat{\mathbf{x}}$ is the input, $\mathbf{c}$ is the convolution coefficients, and $b$ is the bias. The maxpool with the pool size of $3 \times 3$ and stride size of $(2, 2)$ follows for downsampling the size of the output feature maps $\mathbf{y}(i)$ by removing weak features and retaining the largest features. The output of the maxpooling layer is shown as follows:

$$\mathbf{y}_{\max pool}(j) = \max_{p,q} \mathbf{y}[j \times 2 + p + q] \tag{17}$$

where, $\mathbf{y}_{\max pool}(j)$ is the output value of maxpooling, $j$ is the index of the element in the output, and $p, q$ are indices in the pooling window. The rectified linear unit (ReLU) function is used to reduce the computation cost due to fast convergence.

The Conv-unit-2 has the same structure as the Conv-unit-1, the only difference is that the maxpool is replaced by avgpool. The output of the avgpool with pool size of $3 \times 3$ and stride size of $(2, 2)$ is represented as follows:

$$\mathbf{y}_{avgpool}(j) = \frac{1}{9} \sum_{p,q} \mathbf{y}[j \times 2 + p + q] \tag{18}$$

where, $\mathbf{y}_{avgpool}(j)$ is the output value of avgpooling, $j$ is the index of the element in the output, and $p, q$ are indices in the pooling window.

The accuracy of human activity recognition outcomes is dependent mainly upon the features derived from the acquired m-D signatures of the input data. These features indicate the energy level of the micro-movements of the limbs, which are often much smaller in magnitude compared to the overall movements of the entire body. Thus, these features can be seen as weak ones that are easy to filter out through convolutional and pooling layers, which decreases the accuracy of the classification. To address this issue, our suggested solution involves incorporating cross-residual connections within the im-res blocks. By implementing these connections, we aim to preserve and reuse previously meaningful feature maps that may be lost

during the model's extraction backbone flow. The two main innovations of the proposed model show as follows:

Firstly, the convolutional layers of conventional CNNs or DCNNs usually have a size of $3 \times 3$. However, the region containing valuable m-D signatures of various human activities on the input spectrogram is completely different based on the type of activity, distance, and aspect angle of the radar. Therefore, using a filter with a fixed size of $3 \times 3$ will not be effective in extracting features, and determining the ideal filter size for extracting and learning those signatures is a challenge. A larger filter yields a global distribution, whereas a smaller one yields a distribution that is limited to a specific locality. Besides, raising the depth is susceptible to overfitting. The answer to this difficulty is to run different-sized kernels on the same input tensor [34]. Therefore, in the structure of the proposed model, filters of different sizes, including $1 \times 1$, $3 \times 3$, and $5 \times 5$, are intelligently designed to be able to extract the most diverse features, including global and local distribution from the same input feature map. Besides, the design can ensure the best balance between classification accuracy and computational complexity when using more than two conv-unit as well as conventional fixed-size filters in this subblock.

Secondly, the dataset to be classified is spectrograms containing m-D signatures, which might be considered weak features, which are easily filtered through Conv-unit, resulting in a decrement of classification accuracy. To address this issue, cross-residual connections are employed appropriately in im-res blocks to reuse previously useful feature maps, which may be lost when going through the model's extraction core flow. Cross-residual connections have all the main advantages of residual connection (also known as skip connections), such as mitigating the vanishing gradient problem, easier optimization, and improved feature reuse. However, the cross-residual connection makes a longer connection than the residual connection due to cross-connections throughout a $1 \times 1$ conv layer. To be more specific, at the first im-res block, the features obtained from two conv layers with different sizes of $3 \times 3$ and $5 \times 5$, respectively, continue to be accumulated with the features obtained from the immediately preceding norm layer, $N_i$, by residual connection. This accumulation will add weak features that may have been filtered by the two conv layers.

The cross-residual connection (red connection line) is started from the $2^{nd}$ im-res block. Along $N_i$, $\Sigma_i$ will accumulate the $N_{i-1}$ feature from the previous norm layer through a $1 \times 1$ conv layer with a sliding step (2, 2). This will add the weaker features, which represent minor signatures of micro-motion that may have been removed through the layers' previous conv-unit.

The obtained features of the first im-res block, $\Sigma_1$, are expressed as follows:

$$\Sigma_1 = \mathcal{C}_{3 \times 3}(N_i) \oplus \mathcal{C}_{5 \times 5}(N_i) \oplus N_i \tag{19}$$

The obtained features at $\Sigma_i(i = 2 : 7)$ are shown as:

$$\Sigma_i = \mathcal{C}_{3 \times 3}(N_i) \oplus \mathcal{C}_{5 \times 5}(N_i) \oplus N_i \oplus N_{i-1} \tag{20}$$

where, $\oplus$ and $\mathcal{C}$ denote the addition operation and the convolutional operation, respectively; $\Sigma_i$ is the output of the addition layer and the output of the im-res block, also; $N_i$ and $N_{i-1}$ is the input values of im-res block.

Finally, CRCNN finishes with a Con-unit-2, a fully connected (FCN) layer (where the total number of neurons is 11, equal to the number of activity classes in the input dataset), a softmax

layer, and a classification layer. The output of the softmax layer is described as follows:

$$\mathbf{p}_i(\mathbf{x}) = \text{softmax}\{r_i(\mathbf{x})\} = \frac{e^{r_i(\mathbf{x})}}{\sum_j e^{r_j(\mathbf{x})}} \qquad (21)$$

where $r_i(x)$ is the $i^{th}$ element of the output FCN layer. Finally, CRCNN uses the highest probability class to predict the precise action of an incoming signal $x$.

$$\text{Predicted\_action}(\mathbf{x}) = \arg\{\max(\mathbf{p}_i(\mathbf{x}))\} \qquad (22)$$

## Experimental results and discussion

Initially, the noise-added dataset will be used to evaluate the performance of the proposed model based on three aspects: classification accuracy, processing time, and complexity under the influence of changing the number of cross-residual connections and the number of filter channels per convolutional layer. Then, the experimental results of the proposed denoising method are presented in the following section. Finally, the proposed model with fine-tuned hyperparameters is continued to be compared with existing DCNNs and evaluate the effectiveness of the proposed denoising method.

The training and testing operations are performed on a computer setup with an Intel(R) Core™ i5-12400F 2.5 GHz CPU, 32 GB of RAM, and an RTX 3060Ti GPU. For training, a batch size of 16, an initial learning rate of 0.01, and 20 epochs are employed with the optimizer's stochastic gradient descent. Five-fold cross-validation is utilized to test and compare the performance of the proposed model with others.

### CRCNN performance evaluation

**The effect of the number of filters on the performance of the proposed model.** The number of filters in the convolutional layers is modified to 32, 48, 64, 72, and 96 in order to assess the impact of this alteration on the performance of the proposed CRCNN model. The results of assessing model quality are depicted in Fig 8, taking into account three primary factors: accuracy, prediction time, and model complexity. The model utilizing 64 filters demonstrates a notable improvement in average accuracy, reaching 89.11%. This result is considerably better compared to the usage of 32 and 48 filters per convolution layer, which provide accuracies of 82.83% and 84.93%, respectively. In comparison, it is insignificantly lower (only about 0.2%) in the case of using 96 filters. Although the obtained accuracy with 96 filters is marginally greater at 89.32% compared to 64 filters; it comes at the cost of increased prediction time and complexity. The prediction time is 17.39 milliseconds, and the number of learnable parameters is 2.8 million. Hence, the CRCNN model with 64 channels achieves the optimal balance between accuracy and prediction time for the activity classification task.

Furthermore, a similar pattern emerges when the number of C-R blocks is kept at 4 or 6 (as seen in Table 2). As a consequence, using 64 filters for each convolutional layer achieves the optimum trade-off for the proposed model.

**The effect of the filter size in the im-res block on the performance of the proposed model.** Table 3 displays the comparison results for accuracy when changing the size of filters in im-res blocks. The model, which uses 2 parallel convolutional layers with a fixed size of $3 \times 3 // 3 \times 3$, has the fastest prediction time, but the classification accuracy is the lowest (just 85.61%). Models using filters of sizes $3 \times 3 // 7 \times 7$ and $5 \times 5 // 7 \times 7$ achieve accuracy of 88.36% and 87.96%, respectively, but they come at the cost of a sharp increase in learned parameters and prediction time. With a selected size of $3 \times 3 // 5 \times 5$, the CRCNN model achieved the

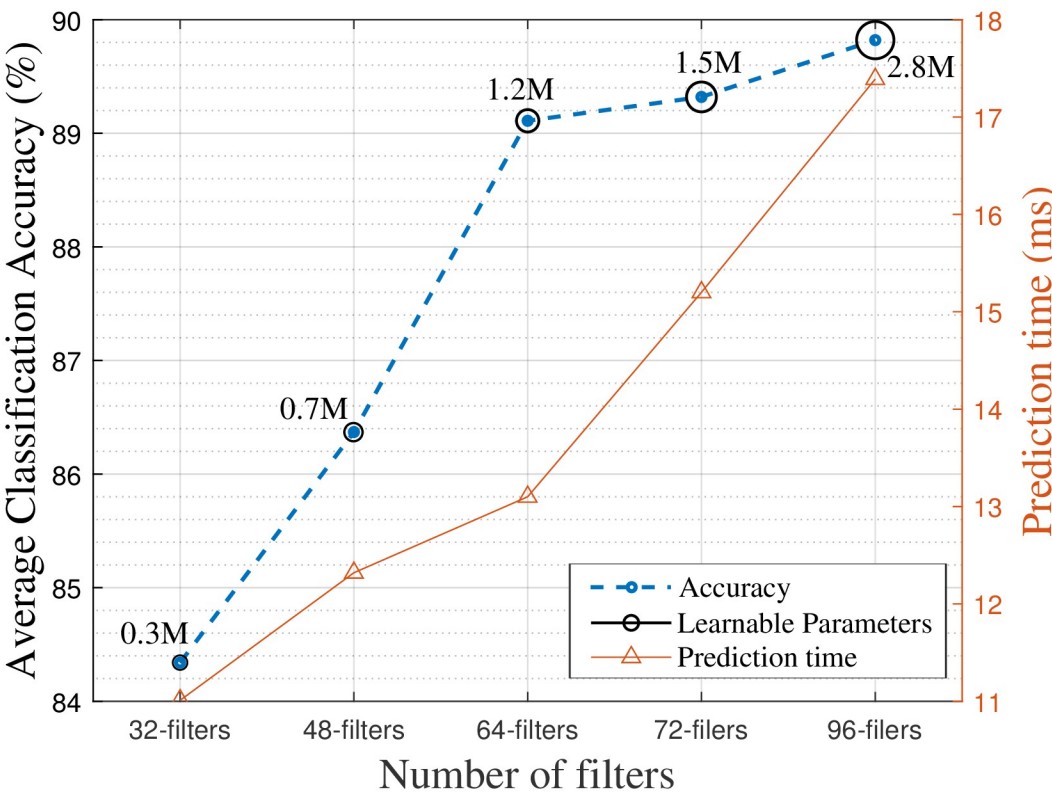

**Fig 8. Performance of CRCNN with different numbers of filter channels.**

highest accuracy (at 89.11%) and a prediction time of 13.1 ms. This result has proven that using filters with different sizes will help extract both local and global features more optimally, and $3 \times 3//5 \times 5$ are the appropriate sizes for the features to be extracted from the input dataset in the proposed CRCNN model.

**The effect of the number of cross-residual connections on the performance of the proposed model.** In this experiment, the influence of the number of cross-residual connections on the performance of CRCNN is assessed. Fig 9 presents the average classification accuracy of CRCNN versus SNR by changing the number of cross-residual connections. A fact that can be observed in Fig 9 is that when the number of connections increases, the accuracy boosts further, which proves that the model is capable of extracting more relevant and useful features. It is argued that when using a large number of connections, the model will get close to the limit of learning efficiency. In this case, the slight increase in accuracy is insufficient to compensate for the quick growth in complexity. Therefore, to compete with other existing models, the model with 6 cross-residual connections and 64 filter channels with the optimal trade-off between accuracy and complexity is proposed for this study.

**Table 2. Average classification accuracy.**

| No. C-R connections | Average classification accuracy (%) | | |
|---|---|---|---|
| | **32 filters** | **64 filters** | **96 filters** |
| 4 | 81.84 | 83.53 | 85.35 |
| 5 | 82.9 | 86.01 | 86.92 |
| 6 | 84.34 | 89.11 | 89.82 |

**Table 3. Compare results with different size filters in the im-res block.**

| Filter size | Accuracy (%) | | | | | | | Para (M) | Train (m) | Pre (ms) |
|---|---|---|---|---|---|---|---|---|---|---|
| | -15dB | -10dB | -5dB | 0dB | 5dB | 10dB | avg | | | |
| 3x3//3x3 | 69.39 | 63.18 | 84.55 | 87.27 | 93.64 | 95.61 | 85.61 | 0.8 | 59 | 12.9 |
| 3x3//5x5 | 74.85 | 86.21 | 87.27 | 91.61 | 96.36 | 98.33 | 89.11 | 1.2 | 61 | 13.1 |
| 3x3//7x7 | 75.76 | 85.3 | 85.61 | 90.61 | 95.3 | 97.58 | 88.36 | 1.9 | 67 | 13.7 |
| 5x5//7x7 | 73.03 | 85.91 | 86.06 | 89.55 | 95.3 | 97.88 | 87.96 | 2.4 | 71 | 14.4 |

**Note**: Para is the learnable parameters, Train is training time and Pre is the prediction time; (M denotes million; m and ms denote minute and millisecond, respectively).

**Performance comparison.** The comparison performance of the proposed CRCNN with six other DCNN models: RepVGG [26], Dop-DenseNet [27], MobileNet-V2 [28], ResNet [29], DINN [14], and DIAT [30] are provided in this section. Initially, the comparison primarily focuses on the added-noise dataset to showcase the exceptional efficacy of the proposed model. Subsequently, these models will be employed to assess the effective of the suggested denoising technique.

- Overview of existing models:

ResNet-18 and MobileNet-V2, which were published in 2016 and 2018, respectively, are two popular models among the six models being compared. These two models are extensively

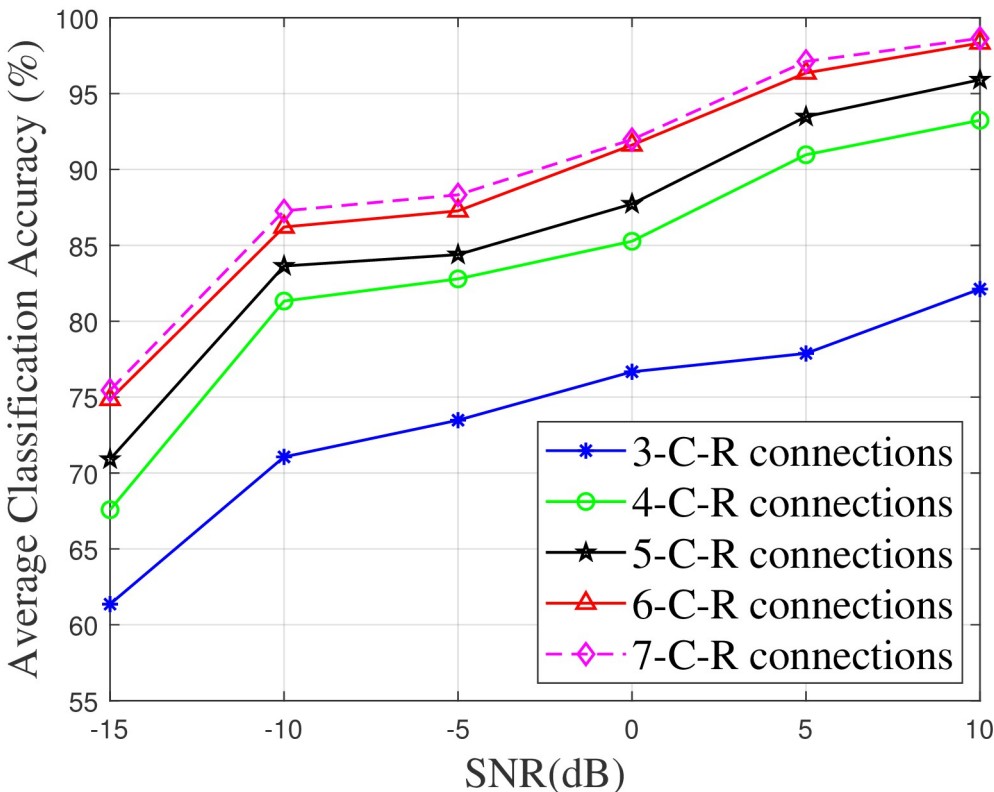

**Fig 9. The average classification accuracy of CRCNN depends on SNR with varied numbers of C-R connections.**

**Table 4. Comparison results of CRCNN with state-of-the-art DCNN models.**

| Classifiers | Acc (%) | Para (M) | Layers | Train (m) | Pre (ms) |
|---|---|---|---|---|---|
| DIAT | 69.9 | 2 | 76 | 146 | 14.6 |
| DINN | 72.5 | 1.7 | 129 | 93 | 13.3 |
| RepVGG | 75.4 | 24.6 | 117 | 86 | 15.4 |
| Dop-DenseNet | 82.1 | 1.7 | 49 | 76 | 14.84 |
| ResNet | 87 | 11.1 | 71 | 72 | 13.4 |
| MobileNet-V2 | 87.9 | 2.2 | 154 | 100 | 14.25 |
| CRCNN | 89.1 | 1.2 | 87 | 61 | 13.1 |

utilized in computer vision using transfer learning because of their ability to achieve high classification accuracy on ImageNet dataset. The two remaining networks, RepVGG and Dop-DenseNet, are state-of-the-art networks introduced in 2021 and 2022, respectively. RepVGG is an improved version of VGG that uses re-parameterization technology to enhance classification accuracy and decrease processing time. Dop-DenseNet is a customized CNN inspired by dense connection, which provides the re-usability of loss features due to forward propagation during training for hand gesture recognition. Moreover, DINN and DIAT are two state-of-the-art methods in radar-based human activity classification, which are published in 2022 and 2023, respectively.

- Comparison results on the noise-added dataset:

Table 4 shows the results of the proposed model compared to other DCNN models. DIAT is a lightweight self-custom network (the learnable parameter is just 2 M), so the extracted features can be confused with the extracted features from the added-noise dataset. This led to the accuracy of DIAT being the lowest (just nearly 70%). Although RepVGG has the most learnable parameters (24.6 M), it provides low classification accuracy (around 76%) and has the longest processing time (approximately 15.4 ms). This is because RepVGG is designed with a fixed filter size of $3 \times 3$, which cannot provide a multi-scale of spatial regions in the m-D spectrogram. In contrast, the number of parameters to be learned by the CRCNN model is the lowest, at around 1.2 M, while the correct classification rate produced by this model is the greatest, at approximately 89.1%. This result can be attributed to the obtained feature diversity and the effective reusability of former features from the previous layers. Furthermore, the prediction time of the suggested network is 13.1 ms, which is comparable to ResNet and DINN and significantly faster than MobileNet-V2 (14.25 ms) and Dop-DenseNet (14.84 ms).

## The proposed denoising method performance evaluation

**Select the optimal range-bin interval based on the minimum entropy criterion.** $\mathbf{r} = \{0, 1, 2, 3\}$ is selected for this experimental. Fig 10 shows the spectrogram with various selected range-bin intervals at 5-dB SNR. Specifically, Fig 10a and 10f depicts the spectrogram of the walking and walking to fall actions at 5-dB SNR when performed STFT on all range-bins of the FFT-range. Fig 10b–10e and 10g–10j illustrate the spectrograms of two actions when performed STFT with different selected range-bin index intervals, $r = 0$, $r = 1$, $r = 2$, and $r = 3$, respectively. Looking specifically at Fig 10b and 10g, with $r = 0$ (corresponding to only one selected $idx_{max}$ range-bin value), the background noise is greatly filtered out, and the m-D signatures become clearer in the noise spectrogram. However, several surrounding areas containing smaller and more detailed m-D signatures still need to be clearly exhibited by the signatures shown in Fig 10c and 10h (dashed area circled in red). For Fig 10d, 10e, 10i and 10j,

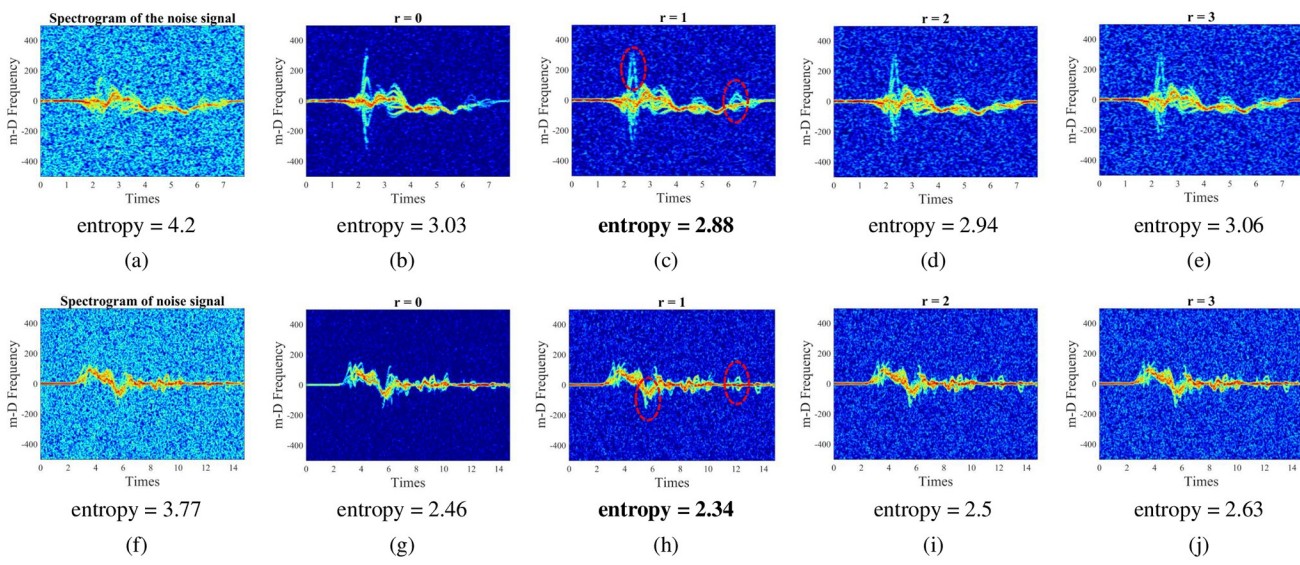

**Fig 10. The spectrogram with various selected range-bin intervals at 5-dB SNR.**

the features are likewise clearly enhanced but are still influenced by background noise. With $r = 1$ (Fig 10c and 10h), the energy level and m-D signatures are represented most clearly and completely for the components of the torso and limbs on the obtained spectrogram image.

Moreover, the results provided in Table 5 also reveal that the achieved minimal entropy value achieved at $r = 1$, corresponding to $r_{opt} = 1$, $range_{opt} = (idx_{max} - 1 : idx_{max} + 1)$, thus, the optimal range-bin interval is 3. A similar result with 0-dB SNR is also presented in Table 5.

**Select the filter cut-threshold value.** The cut-threshold value, $Th$, is slected in this experimental is 2, 3, and 4 respectively. Fig 11 illustrates the spectrogram images of walking and walking to fall actions at -5-dB SNR with various cut-threshold values. Different cut-threshold values produce different m-D signatures in the spectrogram. Fig 11b and 11f displays the spectrogram with a cut-threshold value of 2. Although the m-D signatures are more evident than the initial noise image at -5-dB SNR (Fig 11a and 11e), they are still obscured by background noise. With a cut-threshold value of 4, the background noise has been clearly removed (Fig 11d and 11h); however, the accompanying m-D signatures are also removed, resulting in the loss of derived features used for activity categorization. Fig 11c and 11g shows that the background noise level has been significantly filtered, and the m-D signatures have been preserved practically completely. As a result, with a cut-threshold value of 3, the spectrogram achieves the best balance between denoising and keeping essential m-D signatures that help improve the classification accuracy of various human activities.

**Table 5. Entropy information with different selected range-bin interval.**

| Activities | Entropy information | | | | |
|---|---|---|---|---|---|
| | Noise | $r = 0$ | $r = 1$ | $r = 2$ | $r = 3$ |
| WTF (5dB) | 3.77 | 2.46 | 2.34 | 2.5 | 2.63 |
| W (5dB) | 4.2 | 3.03 | 2.88 | 2.94 | 3.06 |
| WTF (0dB) | 5.37 | 3.46 | 2.96 | 3.22 | 3.42 |
| W (0dB) | 5.4 | 3.61 | 3.3 | 3.37 | 3.66 |

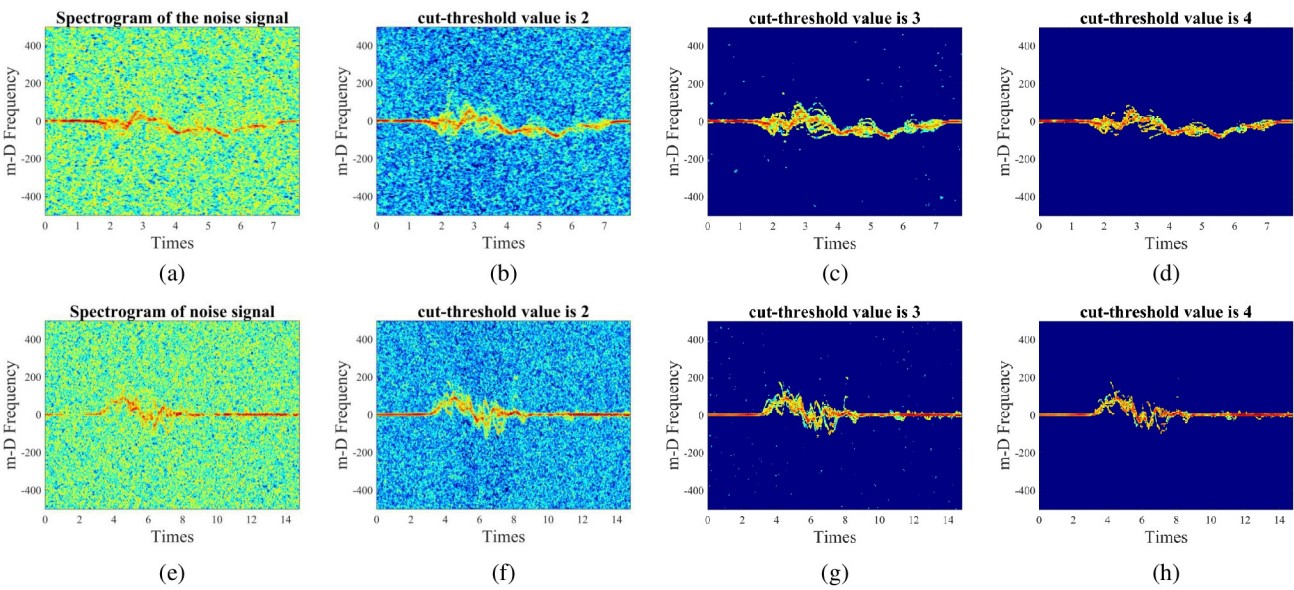

**Fig 11. The spectrogram images of walking and walking to fall actions at -5-dB SNR with different cut-threshold values.** Fig 10a–10d show the walking action. Fig 10e–10h show the walking to fall action.

Table 6 shows the classification accuracy of five CNN models with varying cut-threshold values. The results demonstrate that the models achieve the highest classification accuracy with the selected cut-threshold value of 3. As a result, the chosen cut-threshold value for the proposed denoising technique is 3.

## Evaluate the quality of the proposed approach

**Comparison results of the proposed denoising method with others.** Fig 12 shows the denoising spectrogram of denoising methods. Fig 12a shows spectrogram at different SNR levels; the m-D signatures are almost completely submerged under the noise background at SNR levels less than 0 dB. Although the adaptive threshold (ATh) method has the ability to eliminate noise, the results are not clear and completely fail at SNR levels less than 0 dB (Fig 12b). When background noise is completely removed, the PCC-DT method has a better denoising ability than the ATh method at SNR levels greater than 0 dB, but it comes with the trade-off of significantly losing significant valuable m-D signatures (Fig 12c). This leads to exceptionally high accuracy at 0 dB SNR but gradually decreases at 10 dB SNR. Furthermore, the choice of threshold value and window length of the Hampel filter has a significant impact on the quality

**Table 6. Average classification accuracy with varied cut-threshold values.**

| Classifiers | Accuracy (%) | | |
|---|---|---|---|
| | Cut-threshold value | | |
| | 2 | 3 | 4 |
| RepVGG | 88.46 | 97.07 | 94.6 |
| Dop-DenseNet | 94.37 | 98.52 | 96.11 |
| MobileNet-V2 | 96.62 | 99 | 97.23 |
| ResNet | 93.89 | 98.86 | 96.95 |
| CRCNN | 96.47 | 99.32 | 97.73 |

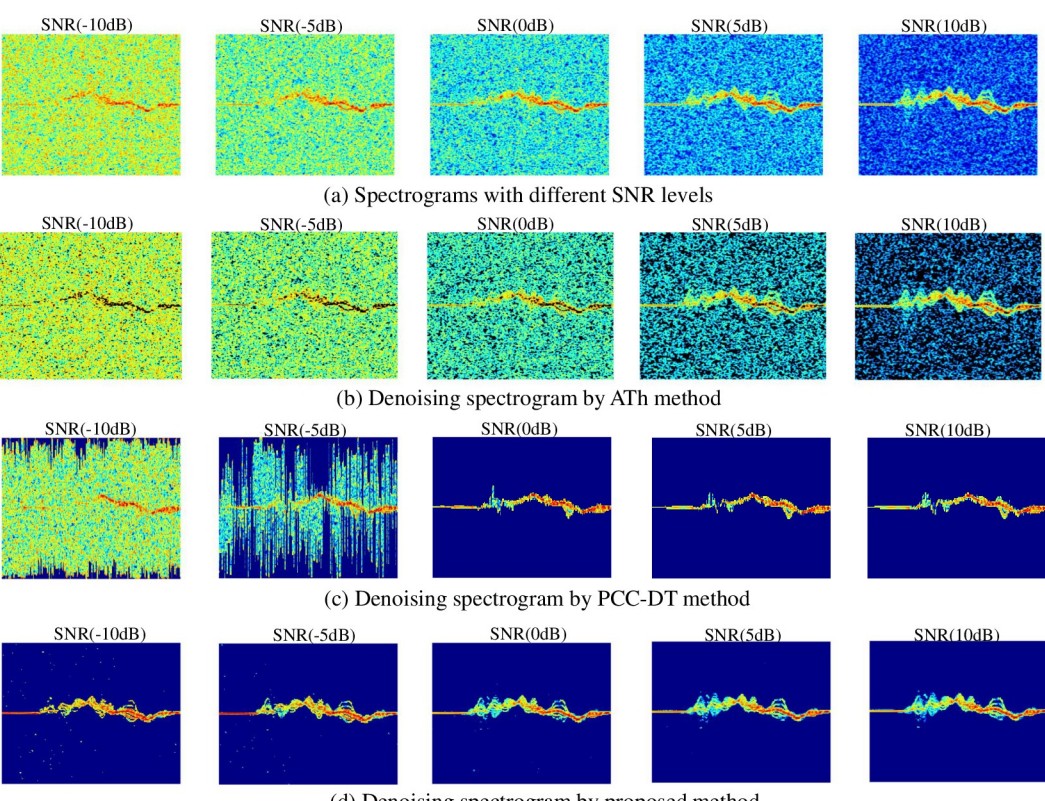

**Fig 12. The spectrograms of different denoising methods for walking action.**

of PCC-DT. These selected values are not suitable for all different SNR levels, so for SNR levels less than 0 dB, this PCC-DT method is not really effective. In addition, the results comparing the classification accuracy of the denoised dataset by the proposed method with other denoising methods by the CRCNN model are shown in Table 7.

Table 7 shows the results of the classification accuracy of the proposed method compared with three other denoising methods, including the ATh, the PCC-DT, and the transfer learning method. The ATh method uses a threshold filter that adapts to different SNR levels to eliminate noise. However, for signals with an SNR level of less than 0 dB, this method is not really effective. A similar result is observed for the PCC-DT method, which uses pattern contour-confined Doppler-time maps to minimize redundant information and remove noise points with high power density. The accuracy was significantly improved at the 0dB-SNR level with

**Table 7. Comparison results of the proposed approach with existing denoising methods.**

| Methods | Accuracy (%) | | | | | | |
|---|---|---|---|---|---|---|---|
| | -15dB | -10dB | -5dB | 0dB | 5dB | 10dB | avg |
| Noise | 74.85 | 86.21 | 87.27 | 91.61 | 96.36 | 98.33 | 89.11 |
| ATh [19] | 70.45 | 83.94 | 85.45 | 93.93 | 97.58 | 98.64 | 88.33 |
| PCC-DT [8] | 75.77 | 79.09 | 84.39 | 96.21 | 97.73 | 98.27 | 88.61 |
| Transfer learning [35] | 73.48 | 85.42 | 86.36 | 90.52 | 94.09 | 97.42 | 87.88 |
| Denoising | 98.33 | 99.09 | 99.39 | 99.55 | 99.7 | 99.85 | 99.32 |

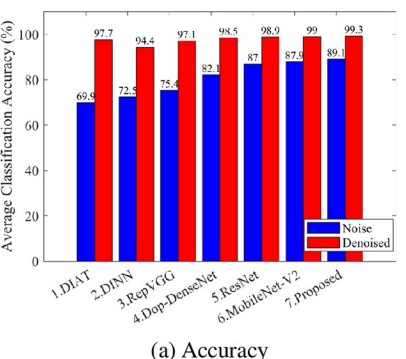
(a) Accuracy

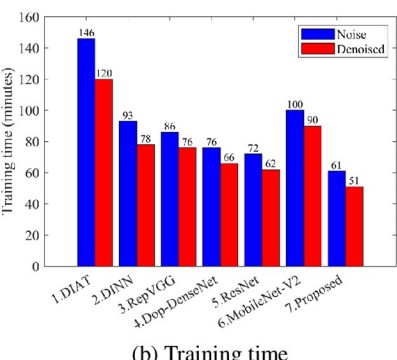
(b) Training time

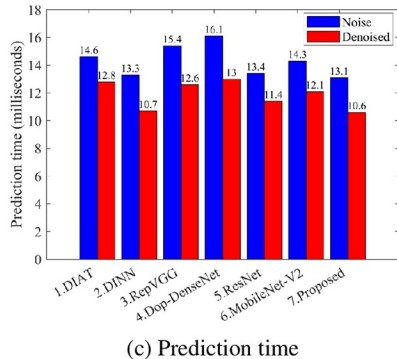
(c) Prediction time

**Fig 13. Performance comparison of different classifiers.**

an accuracy increase of up to 5% but gradually decreased at the 10dB-SNR level because m-D signatures were filtered during the noise denoising process. In contrast to the two methods mentioned above, the transfer learning method uses the pre-trained model to extract features without using a pre-processing step to remove noise; this result shows the accuracy achieved at different SNR levels is the lowest. The proposed method combines denoising in the preprocessing step with the CRCNN model to achieve the best classification results, outperforming the other methods.

**Comparison results of CRCNN with other classifiers.** The results of evaluating the performance of the denoising approach performed on six existing DCNNs and a proposed CRCNN network are shown in Fig 13. Fig 13a demonstrates that the average classification accuracy of all seven classifiers has been significantly increased, which proves that the proposed denoising algorithm really contributes to improving the recognition accuracy of human activities. Specifically, the classification accuracy of all models improved by over 10% for the denoised dataset, in which the suggested CRCNN network reached an average accuracy of more than 99%. Moreover, Fig 13b and 13c shows that using the suggested denoising method significantly cuts down on both the prediction time and the training time of the seven models that were studied. The training time is decreased by roughly 10 minutes, and the prediction time is lowered by about 15% compared to the noise-added dataset.

In addition, three additional important metrics: precision, recall, and F1-score, are also employed to evaluate the denoising effectiveness of the above models (Fig 14). The results indicate the effectiveness in denoising the input datasets of the proposed method.

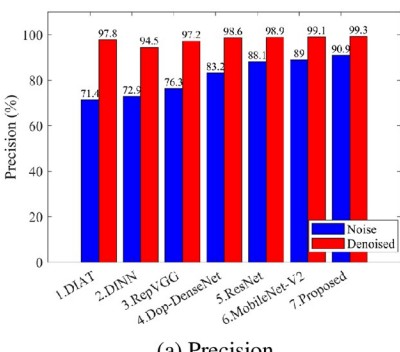
(a) Precision

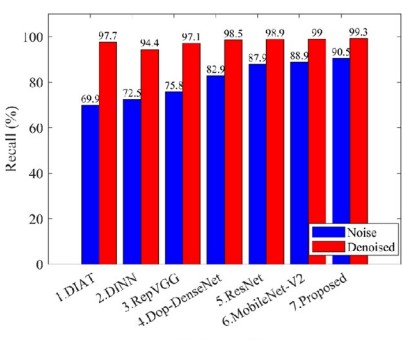
(b) Recall

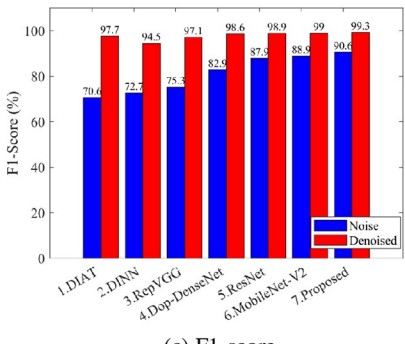
(c) F1-score

**Fig 14. Three metrics: Precision, recall, and F1-socer of different classifiers.**

**Table 8. Performance metrics of different classifiers at various SNR levels on two datasets.**

| Classifiers | Dataset | Accuracy (%) | | | | | | |
|---|---|---|---|---|---|---|---|---|
| | | -15dB | -10dB | -5dB | 0dB | 5dB | 10dB | avg |
| DIAT | Noise | 46.21 | 65.61 | 65.91 | 69.39 | 85.91 | 86.06 | 69.85 |
| | Denoised | 93.64 | 97.27 | 98.33 | 98.64 | 99.09 | 99.24 | 97.7 |
| | **Improved** | **47.43** | **31.66** | **32.42** | **29.25** | **13.18** | **13.18** | **27.85** |
| DINN | Noise | 46.36 | 67.58 | 68.03 | 70.61 | 88.79 | 93.64 | 72.5 |
| | Denoised | 88.18 | 92.58 | 94.24 | 95.3 | 97.73 | 98.48 | 94.42 |
| | **Improved** | **49.4** | **45.75** | **44.85** | **39.7** | **30.91** | **30.76** | **21.92** |
| RepVGG | Noise | 63.18 | 73.33 | 74.55 | 75.61 | 80.91 | 84.7 | 75.38 |
| | Denoised | 94.7 | 96.82 | 97.12 | 97.27 | 97.73 | 98.79 | 97.07 |
| | **Improved** | **31.52** | **23.49** | **22.57** | **21.66** | **16.82** | **14.09** | **21.69** |
| Dop-DenseNet | Noise | 65.91 | 80.45 | 80.76 | 83.64 | 88.33 | 93.33 | 82.07 |
| | Denoised | 97.12 | 98.03 | 98.79 | 98.84 | 99.09 | 99.24 | 98.52 |
| | **Improved** | **31.21** | **17.58** | **18.03** | **15.2** | **10.76** | **5.91** | **16.45** |
| ResNet | Noise | 72.61 | 85.12 | 85.67 | 88.33 | 93.33 | 97.12 | 87.03 |
| | Denoised | 97.27 | 98.79 | 99.09 | 99.24 | 99.35 | 99.39 | 98.86 |
| | **Improved** | **24.66** | **13.67** | **13.42** | **10.91** | **6.02** | **2.27** | **11.83** |
| MobileNet_V2 | Noise | 73.48 | 85.42 | 86.36 | 90.52 | 94.09 | 97.42 | 87.88 |
| | Denoised | 97.42 | 98.94 | 99.24 | 99.39 | 99.48 | 99.55 | 99 |
| | **Improved** | **23.94** | **13.52** | **12.88** | **8.87** | **5.39** | **2.13** | **11.12** |
| CRCNN | Noise | 74.85 | 86.21 | 87.27 | 91.61 | 96.36 | 98.33 | 89.11 |
| | Denoised | 98.33 | 99.09 | 99.39 | 99.55 | 99.7 | 99.85 | 99.32 |
| | **Improved** | **23.48** | **12.88** | **12.12** | **7.94** | **3.34** | **1.52** | **10.21** |

Moreover, the classification accuracy of six existing DCNNs and a proposed CRCNN model evaluated at different SNR levels is shown in detail in Table 8.

According to the experimental results in Table 8, classification accuracy increases as the SNR level rises. This proves that the signal-to-noise ratio directly impacts the classification accuracy of the models. In particular, the classification accuracy on the noise-added dataset of models has a range of about 45% to 75% at an SNR of -15 dB. This range slowly rises to about 85% to 98% at 10 dB-SNR. For the denoised dataset, the accuracy fluctuates from 88% to 98% at -15 dB-SNR and increases from 98% to 99% at 10 dB-SNR. The results show that the denoising method, which uses the appropriate cut-threshold value and the optimal range-bin interval, can effectively eliminate white Gaussian noise. Observing the models in detail, we can see that the improvement gradually decreases as the SNR level increases, which proves that the proposed method is really effective at low SNR levels and still maintains significant improvement at high SNR levels. Furthermore, for both datasets, the proposed CRCNN classification model achieves higher accuracy than the remaining models. Especially when combining the CRCNN and the proposed denoising method, the accuracy reaches over 99% at -10 dB-SNR.

## Performance of the proposed model when changing the distance and aspect angles between object and radar

To evaluate the proposed model's performance when changing the distance and aspect angles between the object and radar, we have built a supplementary dataset consisting of varying distances: 2m, 3m, 5m, and 7m. The aspect angles remain consistent at [0˚, -45˚, 45˚, -90˚, 90˚]. The additional dataset will consist of 13200 samples, which will include 4 distances, 11 actions,

**Table 9. Classification accuracy with various distances.**

| Distances | Accuracy (%) | | | | | | |
|---|---|---|---|---|---|---|---|
| | -15dB | -10dB | -5dB | 0dB | 5dB | 10dB | avg |
| 2m | 83.47 | 92.56 | 93.39 | 97.52 | 98.35 | 99.17 | 94.08 |
| 3m | 81.82 | 91.74 | 94.21 | 96.69 | 97.52 | 98.35 | 93.39 |
| 5m | 78.32 | 87.41 | 91.61 | 93.71 | 96.5 | 97.2 | 90.79 |
| 7m | 77.62 | 86.71 | 90.91 | 92.31 | 95.8 | 96.5 | 89.98 |

10 iterations, 5 aspect angles, and 6 SNR levels. The additional dataset is randomly divided into 80% for training, and the remaining 20% is used for testing. The transfer learning method is applied to the CRCNN model that was trained with the initial dataset.

**Comparison results with various distances.** To evaluate the classification accuracy in detail when the distance between the target and the radar sensor changes, the trained CRCNN model is trained with all distances (2m, 3m, 5m, and 7m) and then tested with each different distance. Table 9 displays the classification accuracy at different distances. The classification accuracy varies with the changing distance between the radar and the target; however, this variation is insignificant (it just fluctuates in the range from 2% to 3%). This result demonstrates the model's capability to accurately categorize targets even when there is a variation in the distance between the target and the radar.

**Comparison results with various aspect angles.** To evaluate the classification accuracy in detail when the aspect angle between the target and the radar sensor changes, the trained CRCNN model is trained with three separate datasets containing different aspect angles and then tested with different aspect angles in turn. In particular, the first dataset only contains data with aspect angle of 0˚ between the person and radar sensor; the second dataset contains aspect angles of 0˚ and ±45˚; and the third dataset contains aspect angles of 0˚, ±45˚, and ±90˚. Fig 15 illustrates the classification accuracy across various aspect angles using three distinct training datasets. The first training dataset, comprising data solely from a 0˚ aspect angle (depicted by the blue line), achieves the highest accuracy at approximately 94% at a 0˚ angle. Conversely, accuracy greatly decreases for other angles due to a lack of training, ranging from 76% to 81%. The second training dataset, which encompasses data from 0˚ and ±45˚ angles (illustrated by the black line), notably enhances accuracy, particularly for the 45˚ and −45˚ angles, with an improvement of up to 10%. However, the observed accuracy at angles 90˚ and −90˚ is still relatively low. Finally, the inclusion of all five aspect angles in the third training dataset (represented by the red line) improves accuracy at 90˚ and −90˚ angles. These findings underscore the impact of aspect angles on classification accuracy and highlight the importance of training datasets.

## Conclusion

In this study, a denoising filter has the task of removing white Gaussian noise from raw radar signals before classifying human activities using CRCNN are proposed to improve the classification accuracy of human activities based on m-D signatures influenced by noise. The denoising algorithm is based on the minimum entropy criterion to determine the optimal range-bin interval of 3; after that, a cut-threshold value of 3 is applied to the filters. The CRCNN model with six C-R connections and 64 filters designated in every convolution layer, along with four state-of-the-art DCNNs, is used to evaluate the denoising performance of the proposed algorithm. Experimental results indicate that the proposed denoising technique has significantly improved classification accuracy by up to 20% when compared to the original noise-added

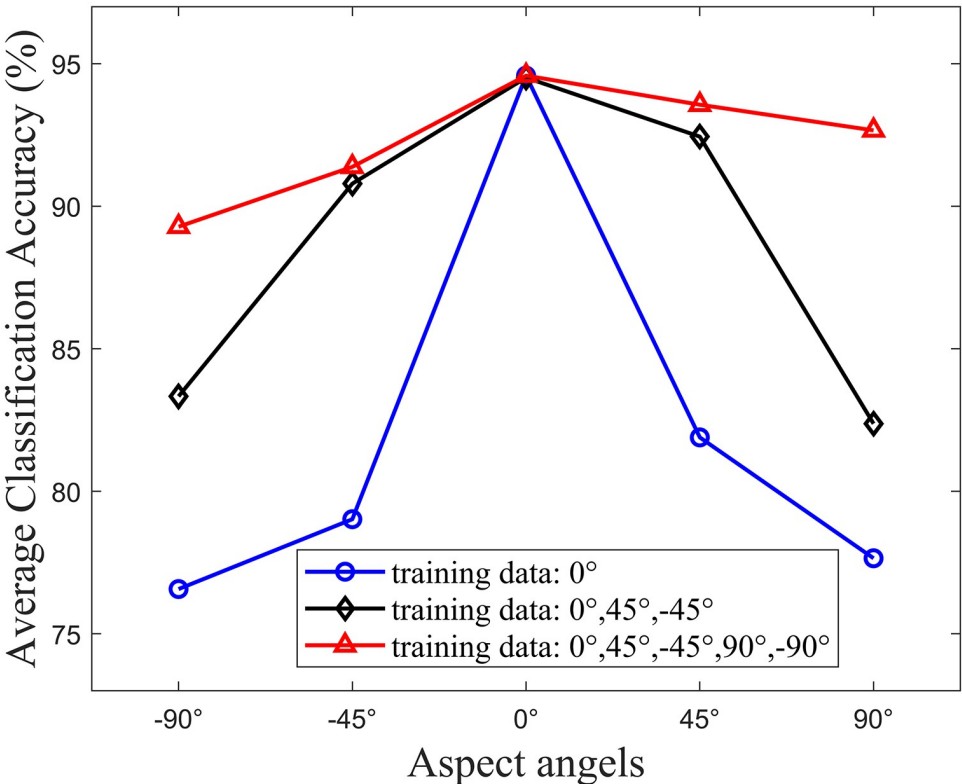

**Fig 15. Classification accuracy with various aspect angles.**

dataset. The suggested CRCNN model exhibits superior accuracy compared to other models, achieving a level above 99% even at a -10 dB noise level. Based on the findings above, future research in this area should focus on addressing the classification of human activities by multiple subjects and considering the impact of different types of noise. The proposed method will be further optimized, verified by experimental measurement, and implemented in a real system for human activity classification applications.

## Author Contributions

**Methodology:** NgocBinh Nguyen, VanNhu Le.

**Software:** VanNhu Le.

**Supervision:** MinhNghia Pham, Van-Sang Doan.

**Validation:** MinhNghia Pham, Van-Sang Doan.

**Writing – original draft:** NgocBinh Nguyen.

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
