## [Decision Letter · Decision Letter 0]

1 Apr 2024

PONE-D-24-04469Improving Human Activity Classification Based on Micro-Doppler Signatures of FMCW Radar with the Effect of NoisePLOS ONE

Dear Dr. Pham,

Thank you for submitting your manuscript to PLOS ONE. After careful consideration, we feel that it has merit but does not fully meet PLOS ONE’s publication criteria as it currently stands. Therefore, we invite you to submit a revised version of the manuscript that addresses the points raised during the review process.

We look forward to receiving your revised manuscript.

Kind regards,

Sushank Chaudhary, Ph.D

Academic Editor

PLOS ONE

Journal Requirements:

3. Thank you for stating the following in your Competing Interests section: "No"

Reviewers' comments:

Reviewer's Responses to Questions

**Comments to the Author**

1. Is the manuscript technically sound, and do the data support the conclusions?

Reviewer #1: Partly

Reviewer #2: Partly

2. Has the statistical analysis been performed appropriately and rigorously? 

Reviewer #1: Yes

Reviewer #2: I Don't Know

3. Have the authors made all data underlying the findings in their manuscript fully available?

Reviewer #1: Yes

Reviewer #2: No

4. Is the manuscript presented in an intelligible fashion and written in standard English?

Reviewer #1: Yes

Reviewer #2: Yes

5. Review Comments to the Author

Reviewer #1: 1. The authors proposed a method to classify human movements using radar spectrograms. In the case of a radar spectrogram, its characteristics change depending on the distance between the person and the radar sensor, the angle between the person and the radar sensor, etc. It is essential to conduct additional experiments considering various conditions.

2. I am curious whether the number of layers, number of filters, and size of filters in the deep learning structure in Figure 6 used by the authors have been optimized. There is a need to justify why the authors set the hyperparameters in the structure as shown.

3. The authors compared classification performance with other deep learning techniques. In addition to classification accuracy, it would be a good idea to compare calculation complexity, calculation time, and time required for data training.

Reviewer #2: The manuscript "Improving Human Activity Classification Based on Micro-Doppler Signatures of FMCW Radar with the Effect of Noise" presents a novel approach for enhancing the accuracy of human activity classification using micro-Doppler signatures from FMCW radar in noisy environments. However i have few questions:

1. Does the abstract clearly summarize the main contributions and findings of the study?

2. Is the significance of the proposed approach in the context of existing challenges in human activity classification using radar sensors adequately highlighted?

3. How does the proposed approach differ from existing methods in handling environmental noise in radar-based human activity classification?

4. What are the specific limitations of visual perception-based methodologies that the proposed radar sensor solution aims to overcome?

5. How is the optimal range-bin interval determined using minimum entropy information, and what is the theoretical basis for this approach?

6. Can you provide more details on the architecture of the Cross-Residual Convolutional Neural Network (CRCNN) and the rationale behind its design choices?

7. How do the adaptable cross-residual connections in the CRCNN model contribute to its performance in classifying activities based on micro-Doppler signatures?

What are the advantages of using different filter sizes in the CRCNN model, and how do they affect feature extraction?

8. How does the proposed denoising technique specifically enhance the classification accuracy, and what are the key factors influencing its effectiveness?

9. Can you provide a more detailed analysis of the experimental results, including the impact of different noise levels on the classification accuracy?

10. How do the findings of this study compare with other state-of-the-art methods in radar-based human activity classification?

11. What are the potential limitations of the proposed approach, and how might they be addressed in future research?

12. What are the main implications of the study's findings for the field of human activity classification using radar sensors?

13. What future directions for research are suggested by the results of this study?

6. PLOS authors have the option to publish the peer review history of their article (what does this mean?). If published, this will include your full peer review and any attached files.

Reviewer #1: No

Reviewer #2: **Yes: **ABHISHEK SHARMA

---

## [Author Response · Author response to Decision Letter 0]

24 May 2024

The content of responses to reviewers' comments is attached in the file "Response to Reviewers".

---

## [Decision Letter · Decision Letter 1]

17 Jul 2024

Improving Human Activity Classification Based on Micro-Doppler Signatures of FMCW Radar with the Effect of Noise

PONE-D-24-04469R1

Dear Dr. Pham,

We’re pleased to inform you that your manuscript has been judged scientifically suitable for publication and will be formally accepted for publication once it meets all outstanding technical requirements.

Kind regards,

Sushank Chaudhary, Ph.D

Academic Editor

PLOS ONE

Additional Editor Comments (optional):

Reviewers' comments:

Reviewer's Responses to Questions

**Comments to the Author**

1. If the authors have adequately addressed your comments raised in a previous round of review and you feel that this manuscript is now acceptable for publication, you may indicate that here to bypass the “Comments to the Author” section, enter your conflict of interest statement in the “Confidential to Editor” section, and submit your "Accept" recommendation.

Reviewer #1: All comments have been addressed

Reviewer #2: All comments have been addressed

2. Is the manuscript technically sound, and do the data support the conclusions?

Reviewer #1: Yes

Reviewer #2: Yes

3. Has the statistical analysis been performed appropriately and rigorously? 

Reviewer #1: Yes

Reviewer #2: Yes

4. Have the authors made all data underlying the findings in their manuscript fully available?

Reviewer #1: Yes

Reviewer #2: Yes

5. Is the manuscript presented in an intelligible fashion and written in standard English?

Reviewer #1: Yes

Reviewer #2: Yes

6. Review Comments to the Author

Reviewer #1: The authors conducted additional experiments and finalized their responses to my comments. There are no more comments.

Reviewer #2: Thank you for addressing the concerns. The manuscript now appears more technically sound and is easier to understand.

7. PLOS authors have the option to publish the peer review history of their article (what does this mean?). If published, this will include your full peer review and any attached files.

Reviewer #1: No

Reviewer #2: No

---

## [Editor Report · Acceptance letter]

22 Jul 2024

PONE-D-24-04469R1 

PLOS ONE

Dear Dr. Pham, 

I'm pleased to inform you that your manuscript has been deemed suitable for publication in PLOS ONE. Congratulations! Your manuscript is now being handed over to our production team.

Kind regards, 

on behalf of

Prof. Sushank Chaudhary 

Academic Editor

PLOS ONE